# Social Inclusion and Physical Activity in Ciclovía Recreativa Programs in Latin America

**DOI:** 10.3390/ijerph18020655

**Published:** 2021-01-14

**Authors:** Carlos Mejia-Arbelaez, Olga L. Sarmiento, Rodrigo Mora Vega, Mónica Flores Castillo, Ricardo Truffello, Lina Martínez, Catalina Medina, Oscar Guaje, José David Pinzón Ortiz, Andres F Useche, David Rojas-Rueda, Xavier Delclòs-Alió

**Affiliations:** 1School of Medicine, Universidad de Los Andes, Bogotá, Carrera 1, n°18A-12, Bogotá 111711, Colombia; cm.mejia134@uniandes.edu.co (C.M.-A.); josed.pinzon@gmail.com (J.D.P.O.); 2Faculty of Architecture and Urbanism, Universidad de Chile, Av. Portugal 84, Santiago de Chile 8331051, Chile; rodrigomora@uchile.cl; 3Centro de Desarrollo Urbano Sustentable, Observatorio de Ciudades UC, Los Navegantes 1963, Providencia, Santiago de Chile 7520246, Chile; maflore1@uc.cl (M.F.C.); rtruffel@uc.cl (R.T.); 4Centro de Desarrollo Urbano Sustentable, Los Navegantes 1963, Providencia, Santiago de Chile 7520246, Chile; 5Faculty of Architecture, Design and Urban Studies, Pontificia Universidad Católica de Chile, El Comendador 1916, Providencia, Santiago de Chile 7520246, Chile; 6Universidad ICESI, Observatorio de Políticas Públicas (POLIS), Calle 18 #122-135, Santiago de Cali 760031, Colombia; lmmartinez@icesi.edu.co; 7Center for Nutrition and Health Research, Department of Physical Activity and Healthy Lifestyles, National Institute of Public Health, 7a. Cerrada de Fray Pedro de Gante #50, Col. Sección XVI Tlalpan, Mexico City 14080, Mexico; catalina.medina@insp.mx; 8Department of Industrial Engineering, School of Engineering, Universidad de Los Andes, Carrera 1, n°18A-12, Bogotá 111711, Colombia; oo.guaje10@uniandes.edu.co (O.G.); af.useche10@uniandes.edu.co (A.F.U.); 9Environmental and Radiological Health Sciences, Colorado State University, Environmental Health Building, 1681 Campus Delivery, Fort Collins, CO 80523, USA; david.rojas@colostate.edu; 10Institute of Urban and Regional Development, University of California, Berkeley, 316E Wurster, Berkeley, CA 94720-1870, USA; xavidelclos@berkeley.edu

**Keywords:** urban segregation, social inclusion, physical activity, ciclovía program, open streets, cross-sectional study

## Abstract

Ciclovía Recreativa is a program in which streets are closed off to automobiles so that people have a safe and inclusive space for recreation and for being physically active. The study aims were: (1) to compare participant’s spatial trajectories in four Ciclovía Recreativa programs in Latin America (Bogotá, Mexico City, Santiago de Cali, and Santiago de Chile) according to socioeconomic characteristics and urban segregation of these cities; and (2) to assess the relationship between participants’ physical activity (PA) levels and sociodemographic characteristics. We harmonized data of cross-sectional studies including 3282 adults collected between 2015 and 2019. We found the highest mobility for recreation in Bogotá, followed closely by Santiago de Cali. In these two cities, the maximum SES (socioeconomic status) percentile differences between the neighborhood of origin and the neighborhoods visited as part of the Ciclovía use were 33.58 (*p*-value < 0.001) and 30.38 (*p*-value < 0.001), respectively, indicating that in these two cities, participants were more likely to visit higher or lower SES neighborhoods than their average SES-of-neighborhood origin. By contrast, participants from Mexico City and Santiago de Chile were more likely to stay in geographic units similar to their average SES-of-origin, having lower overall mobility during leisure time: maximum SES percentile difference 1.55 (*p*-value < 0.001) and −0.91 (*p*-value 0.001), respectively. PA levels of participants did not differ by sex or SES. Our results suggest that Ciclovía can be a socially inclusive program in highly unequal and segregated urban environments, which provides a space for PA whilefacilitat physical proximity, exposure to new communities and environments, and interactions between different socioeconomic groups.

## 1. Introduction

In the 21st century, cities around the world are facing the consequences of rapid, unplanned urbanization [1]. Particularly, Latin America is a region with high rates of urbanization, inequality, and urban segregation [2].

Currently, about 80% of the Latin American population lives in cities, towns, and other urban settlements [3], with proliferation of informal housing settlements at the periphery of cities and creating the traditionally recognized spatial structure of Latin American cities: lower-income groups tend to occupy peripheral, homogenous, and poorly-serviced areas; meanwhile, high-income groups tend to be clustered in one area of the city, with little or no interaction with their counterparts [4]. Such segregation has been associated with existing violence and conflicts among groups [5], as well as perpetuating rifts in societies and exacerbating inequalities through unequal service provision and disparities in environmental conditions [6,7]. Additionally, socioeconomic inequality extends to every single aspect of life, from distribution of income, land, and other assets to access to health services, public spaces, and green zones, education, justice, and political voice [8,9,10], factors that are all related to unhealthy and unsustainable development [11].

Urban segregation occurs either by a self-selection process (in the case of affluent groups) or through the inability to pay for desirable locations in a city (vulnerable communities) [12,13,14]. Moreover, it occurs on different scales and forms [12,13,14]. It can be a consequence of a large-scale geographical separation between social groups, potentially promoted by governmental policies [15], or by operation of market forces [16], either manifested by the construction of new developments in the outskirts of cities [17] or large-scale renewal initiatives resulting in gentrification trends [5]. Urban segregation can be enforced by physical barriers between different parts of the city (e.g., rivers, ravines, or highways) [18] or symbolic barriers that divide proximal neighborhoods such as fences, walls, and controlled entrances [19,20,21].

In addition, urban segregation affects people’s lives in various ways. It preserves and exacerbates existing violence and conflicts among groups [5]. Urban segregation reduces access to parks and public transport and increases travel time for low-income groups [6], contributing to urban inequalities and decreased opportunities [15,22,23]. Additionally, because segregation affects how urban commons goods are distributed, it determines to a great extent people’s quality of life [24,25].

In Latin America, urban segregation has historically shaped the way cities have grown [15,19,26], permitting high-income groups to self-isolate in cities. It has been argued that this process is caused by a combination of neo-liberal policies coupled with deep-rooted historical factors. Since the 1990s, however, international agencies have established the need to overcome urban segregation as a crucial step in pursuing more equitable societies [15,19,26]. As a result, Latin-American governments have tried to mitigate urban segregation to promote social integration [15,27,28].

Within this context, innovative public space and transport programs and policies have emerged seeking to activate urban spaces by enabling neighbors and families to socialize and do physical activity (PA). One of these programs is the Ciclovía Recreativa, in which streets are temporarily opened to persons and closed off to automobiles so that people have a safe and inclusive space for recreation and for being physically active [29]. By providing an alternative use of public space, the Ciclovía Recreativa functions as a socially inclusive program that provides a weekly recreational alternative to different socioeconomic groups, often to those with the poorest access to green spaces and recreational premises in their neighborhoods. In this context, Ciclovía Recreativa transforms streets (albeit temporarily) into democratic and inclusive spaces where people can interact and use them on their own [29,30].

There are around 497 Ciclovía programs in 27 countries globally, with Latin America being the region with the greatest number of regular programs [30,31]. Evidence coming from programs of Bogotá [32,33], Mexico City [34], Santiago de Cali [35], and Santiago de Chile [27] shows that these programs promote healthy lifestyles [30,33,36,37], reduce exposure to air pollution and street noise [38,39] and enhance inhabitant’s quality of life [33,40]. Of particular interest is the role of the Ciclovía in promoting PA, especially in Latin America and the Caribbean region, the region with the overall highest prevalence of insufficient PA worldwide (43.7%) [41]. However, there is limited evidence about to what extent multiple Ciclovía Recreativa programs provide leisure opportunities to spatially segregated populations. A previous study in Bogotá shows urban inequities in the distribution of the Ciclovía program, but the trajectories of the participants have not been evaluated, and studies including other cities have not been conducted [42]. We hypothesized that these programs allow people to navigate through multiple urban spaces during recreation, thereby helping citizens to overcome pervasive patterns of urban segregation.

A better understanding of the potential of the Ciclovía as a socially inclusive program requires a dynamic measure of segregation that, in contrast to the traditional static and spatial segregation measures, recognizes and assesses segregation as a dynamic process that relies on individuals’ daily life routines, as well as on the different ways they or social groups use urban space [43,44]. In this respect, a segregation measure should consider the dynamism of the “flow” of people in the Ciclovía rather than a place-based approach. Therefore, the aim of this study was twofold: firstly, to compare participant’s spatial trajectories in four Ciclovía Recreativa programs in Latin America (Bogotá, Mexico City, Santiago de Cali, and Santiago de Chile) according to socioeconomic characteristics and urban segregation of these cities; and secondly, to assess the relationship between participants’ PA levels and sociodemographic characteristics. This study will provide scientific evidence to the local public policy decision-makers on the potential of these programs to improve social inclusion in cities, and to promote PA in cities of Latin America.

## 2. Materials and Methods

### 2.1. Study Settings

This cross-sectional study included data from Ciclovía programs in four Latin American cities: Bogotá and Santiago de Cali, Colombia; Mexico City, Mexico; and Santiago de Chile, Chile. These cities are characterized by having large populations, being highly dense, unequal, violent, and fragmented, and having limited access to recreational resources, including parks (Table 1).

### 2.2. Characteristics of the Ciclovía Recreativa Programs

#### 2.2.1. Bogotá: Ciclovía

Ciclovía of Bogotá is the program with the largest number of closed street kilometers in the world. It is a multisectoral program inaugurated in 1974 and coordinated by the District Institute of Sports and Recreation. It comprises a 127.69 km circuit that crosses 18 of 20 localities of Bogotá [45]. The seven-hour events occur on Sundays and holidays, with about 66–72 events per year and an estimate of 600,000 to 1,750,000 participants each event [30,46] (Table 1).

#### 2.2.2. Mexico City: Muévete en Bici (MEB)

This program was inaugurated in 2007 by the Ministry of Environment in partnership with 20 private and public organizations of Mexico City. It consists of 55 km of interconnected streets that are closed from 8:00 a.m. to 2:00 p.m. on the first three Sundays of each month, with about 37 events per year. On average, an estimate of 21 thousand people attends the program every Sunday [34,47] (Table 1).

#### 2.2.3. Santiago de Cali: Ciclovida

The Ciclovida, inaugurated in 1996, is a program of the Mayor of Santiago de Cali, led by Cali’s Secretariat of Sport and Recreation. The circuit is segmented in one central lane that goes from north to south, and seven community lanes, for a total route length of 60 km. These lanes are opened for city dwellers every Sunday from 8:00 a.m. to 1:00 p.m. and every Thursday from 8:00 p.m. to 10:00 p.m., with about 93 events per year, and average participation of approximately 30,000 people every Sunday [35,48] (Table 1).

#### 2.2.4. Santiago de Chile: CicloRecreoVía

CicloRecreoVía is run by the private consultancy firm Geomás. It was inaugurated in 2006 with one circuit of 5 km that progressively was expanded to 11 circuits, for a total of 38 km. It is composed of seven routes primarily located in Santiago’s affluent north-east area and the historical center, with only one circuit located near middle and lower-middle-income neighborhoods. The events take place from 9:00 a.m. to 2:00 p.m. on Sundays, with about 51 events per year and average participation of 40,000 people every Sunday [27,49] (Table 1).

### 2.3. The Ciclovía Recreativa Program Surveys

This study included data from 3282 individuals collected between 2015 and 2019 (Bogotá, N = 1001; Mexico City, N = 721, Santiago de Cali, N = 1159; Santiago de Chile, N = 401). In each of the cities, trained interviewers conducted intercept surveys for which participants were selected systematically. Each participant took a questionnaire that lasted between 10–15 min to be completed, and it was done using pen-and-paper in Bogotá, Santiago de Cali, and Santiago de Chile; or the Survey Monkey platform [50] on a cell phone in Mexico City. Surveys included questions about sociodemographic, participant’s health characteristics and PA behaviors, program use, and participants’ safety perceptions. In Mexico City and Santiago de Chile, participants provided their written informed consent before participating, whereas in Bogotá and Santiago de Cali, participants gave their verbal consent prior to participation. For these surveys, interviewers received training that varied by city. In Bogotá, training was for 10 h. for two days, including a class and a practical workshop with specific scenarios. In Mexico City and Santiago de Cali, interviewers piloted the questionnaire two and five times, respectively before its implementation. In Santiago de Chile, students were trained to conduct the interviews and a pilot study was conducted to adjust the questionnaire before its implementation.

We developed a process for harmonization of the surveys following the SALURBAL (Salud Urbana en America Latina/Urban Health in Latin America) protocol [51]. First, we identified and collated questions and responses of each survey with attention to skip patterns and respondent universe. Second, we reviewed surveys conducted by the World Health Organization for standard variable definitions as well as harmonization approaches proposed by other projects [41]. Third, we created variable definitions and response categories with attention to differences in wording across countries. Fourth, we applied the harmonization and revised the protocol based on descriptive statistics of initial harmonized variables. We harmonized and developed a data structure that accommodates information available at the individual and city levels. The process was guided by the principle that pragmatic albeit imperfect definitions would be necessary to advance in the knowledge of the potential benefits of the Ciclovías.

Ethical approval was obtained in each country from the principal investigator’s institutional review board. Universidad de los Andes-Colombia Research Ethics Committee in Bogotá (act no. 691–2017), The National Public Health Institute Ethics Review Board of Mexico in Mexico City (act no. 1703), Universidad ICESI Research Ethics Committee in Santiago de Cali (act no. 076), and the Pontificia Universidad Católica de Chile Ethics Committee in Santiago de Chile, approved the study in each of the cities where the study took place. A secondary analysis was approved by Universidad de los Andes-Colombia Research Ethics Committee (act no. 1017).

#### 2.3.1. Sociodemographic Characteristics

Sociodemographic characteristics included sex, birthdate (in Mexico City, Santiago de Cali, and Santiago de Chile), age (in Bogotá), marital status (single, married, widowed, divorced, separated), last grade of school completed (primary, secondary/high school, college, master degree or higher), car availability in the household (yes/no), home address, including their zip code (only in Mexico City and Santiago de Chile surveys), and participant’s neighborhood socioeconomic status (SES). We computed age from the date of birth to the observation date and classified participants into the following categories: 18–29 years, 30–49 years, or ≥50 years. Marital status was dichotomized into single or living with a partner. In each of the cities included in the study, the participant’s neighborhood was defined as the smallest geographic unit for which there were available data: in Bogotá and Santiago de Cali, we employed the neighborhood level, while in Mexico City and Santiago de Chile, we employed locality and census zones level, respectively. SES variables from each city corresponds to indicators used for statistics in each city. In Bogotá and Santiago de Cali, SES was determined by using the classification from the National Administrative Department of Statistics (DANE, by its initials in Spanish), which has six categories based on physical characteristics of the household and neighborhood area (SES category 1 corresponds to the poorest and category 6 to the richest) [52]. In Mexico City, zip code was used to estimate SES based on the National Institute of Statistics, Geography, and Informatics database (by its acronym in Spanish: INEGI) [53]. In this city, participants were classified using the following categories: low, middle, and high level. In Santiago de Chile, SES was determined based on the socioeconomic index created by the Chilean National Automotive Association and the Chilean Association of Market Researchers (by its acronym in Spanish: ANAC and AIM, respectively) [54,55]. This later index classifies people based on the education level of the main supporter of home and a battery of 10 goods as follows: E (Very low), D (Low), C3 (Middle low), C2 (Middle high), and ABC1 (High) [54,55,56]. To have comparable groups among cities on regard SES, categories 1 and 2, 3 and 4, 5, and 6 from DANE classification were merged into three categories: low, middle, and high level, respectively. In Santiago de Chile, we merged categories E and D into the low level, and C3, and C2 into the middle level.

#### 2.3.2. Health Characteristics

To evaluate self-assessed health status, participants were asked: ‘Compared to people your age, how would you rate your health over the past month?” with the possible choices being “very good” (1), “good” (2), “not good, not bad” (3), “bad” (4), or “very bad” (5). The last three outcomes were then grouped into a “fair” health category. For all cities, except for Santiago de Chile, participants reported their height and weight. Body Mass Index (BMI) was calculated using the WHO criteria [57].

#### 2.3.3. Physical Activity Behaviors

Using the question regarding time participants spent at the Ciclovía program doing PA, we classified participants into meeting or not PA recommendations during the program. Additionally, weekly PA levels during transport and leisure time were determined in Bogotá and Mexico City employing the International Physical Activity Questionnaire (IPAQ) [58] and the Global Physical Activity Questionnaire (GPAQ) [59], respectively. The IPAQ has been validated in Latin American countries [60,61] and the GPAQ has been validated in 10 countries [62]. Participants were classified into the following categories: Meeting PA recommendations at transportation or leisure time; meeting PA recommendations in leisure time (LTPA); and meeting PA recommendations by walking and cycling for transport. We classified the variables regarding meeting PA recommendations as yes or no, based on the World Health Organization (WHO) PA recommendations for adults aged 18–64: ≥150 min/week of moderate-intensity aerobic PA, ≥75 min/week of vigorous-intensity aerobic PA, or an equivalent combination [63].

#### 2.3.4. Program Use

To assess Ciclovía participation patterns, we asked participants: “What is the main activity you perform during the program?”, “How many hours do you usually spend in the program?”, and “How often do you attend Ciclovía?”. Ciclovía users were characterized according to the type of activity they performed during the program (cycling, rollerblading, walking, running/jogging, other), time spent at Ciclovía (<3 h, 3–4 h, ≥ 4 h), and the frequency of participation (every Sunday, two/three times per month, once per month, and at least once a year). Questions about the type of activities they normally do on Sundays if they are not attending the program (sedentary activities, active activities, or very active activities), their reasons to attend the program (share with family/friends, used for recreation/PA, health benefits, because it is free, attend to free exercise classes, other), and whether or not they attend the program accompanied (i.e., family member, co-workers, partner, neighbor, classmate, friend) were also included. People accompanied by at the program variable was coded as “came alone” or “came with another person”.

#### 2.3.5. Participants’ Perceptions

We included an item related to participants’ safety perceptions at the program. This question was scored using a Likert scale from 1 to 5, with 1 indicating that the respondent strongly disagreed with the positive safety statement and 5 indicating that the respondent strongly agreed. This variable was then classified into “unsafe”, “safe”, “neither safe nor unsafe” categories.

### 2.4. Urban Segregation

We defined urban segregation as the degree to which people from two or more SES live separately from one another at the different spatial units of the city [64].

#### 2.4.1. Urban Segregation Index

The segregation of groups in urban space manifests itself in several ways, and each corresponds to a different aspect of the same phenomenon [64,65]. The literature distinguishes numerous dimensions and indices to measure urban segregation, each of them provides a way to describe and compare the distribution of population groups—defined by age, ethnic origin, country of birth, income—across a metropolitan area [66]. This study employed the dimension of evenness defined by Massey and Denton in 1988 [67], which refers to the differential distribution of social groups among areal units in a city or, in other words, how “even” the distribution of the different groups of the population across spatial units within the city is. Particularly, for each of the cities included in the study, we calculated the Theil index or the entropy index/diversity index, one of the most common indices employed as a measure of evenness. By using this, we characterized the extent to which individuals of different SES (i.e., low, middle, and high SES) are evenly distributed throughout geographic units (i.e., neighborhoods, localities, census zones). To calculate this index we used the application developed by Philippe Apparicio et al. [66]—Geo-Segregation Analyzer. The entropy index ranges between 0 and 1 and allowed us to identify geographic units that are completely homogenous (inhabited by only one SES group, H = 0) or maximally diversified (all SES groups are equal in size, H = 1) [67]. This index is defined by the following formulas [67,68,69].

For each geographic unit i (neighborhood, locality, or census zone), located within a metropolitan area, the entropy score (E_i_) is defined as
(1)$$E_{i}=\overset{r}{\underset{r=1}{\sum }}\left(\pi_{ri}\right)\log\left[\frac{1}{\pi_{ri}}\right]$$
where π*_ri_* refers to a particular SES proportion (i.e., low, middle, and high SES) of the total metropolitan population in that specific geographic unit (i) and “r” indexes the socioeconomic groups (e.g., SES levels) in a specific population [67,68,69]. For a single geographic unit (i), the entropy score (H), measures the extent to which the geographic unit’s entropy (E_i_) is reduced below the metropolitan’s entropy (E). In other words, the entropy index (H) is the weighted average deviation of each geographic unit’s entropy ((E_i_) from the metropolitan-wide entropy (E), expressed as a fraction of the metropolitan area’s total entropy
(2)$$H=\overset{n}{\underset{i=1}{\sum }}\left[\frac{t_{i}\left(E-E_{i}\right)}{E*T}\right]$$
where *t_i_* refers to the total population of the geographic unit (i), T is the metropolitan area population, n is the number of geographic units, and E_i_ and E represent geographic unit i’s and metropolitan area entropy, respectively. The entropy index is at its maximum when urban segregation is at its minimum, meaning that SESs are evenly distributed across geographic units or that all geographic units of the city have an equal proportion of the population belonging to the different SES. In contrast, the index is minimized—and urban segregation is maximized—when any pair of individuals from two different SESs does not inhabit the same geographic unit [64].

For this study, the geographic units, as well as the SES proxy’s employed to obtain the entropy index varied by city, as a result of each’s country available information. As geographic units in Bogotá and Santiago de Cali, we employed the neighborhood-level, the locality-level in Mexico City, and the census zone level for Santiago de Chile. In Bogotá and Santiago de Cali, SES was defined according to the DANE [52]. In Mexico City, we employed as SES’s proxy the marginalization index, an aggregate measure of social deprivation elaborated by the National Population Council of Mexico (CONAPO, by its acronym in Spanish) [70]. It ranks states, municipalities, and localities in Mexico into the following categories: very high, high, moderate, low, and very low [71]. Finally, in Santiago de Chile, we used as SES’s proxy the index institutionalized by the ANAC and AIM [54,55,56].

#### 2.4.2. Participants’ Trajectories Through the Ciclovía Program

We calculated the participants’ trajectories through the Ciclovía program from the participant’s origin point to the participant’s destination point. Origin and departure points by study cite are presented in Table 2. Trajectories were drawn using ArcGIS^®^ software (ArcGIS 10.7.1; ESRI Inc., Redlands, CA, USA) based on the shortest path distance from the origin to destination points through the street network. Datasets included in the analysis were: an origin-destination matrix, the spatialized trajectories of each journey through each city, and an income or socioeconomic proxy variable for each city referenced at the smallest available geographic unit.

#### 2.4.3. Average SES per Traveled Kilometer

In addition to participants’ trajectories through the Ciclovía program, we calculated the average SES per traveled kilometer of all participants’ trajectories, that is the changes in the socioeconomic characteristics of urban environments during users’ trajectories (to reach the Ciclovías and through the Ciclovía route). We first divided the trajectories every 500 m and spatially join the SES information at the smallest available geographic unit. Then, we grouped all participants by SES-of-origin (i.e., low, middle, and high) and calculated an average socioeconomic score and variance for every 500-m segment of the total traveled distance, which was measured starting from the origin point of all journeys by SES-of-origin category. This estimation allowed us to assess whether the participant’s trajectories transverse different socioeconomic environments.

### 2.5. Statistical Analyses

First, we compiled and harmonized common variables in each of the four cities. Then descriptive statistics (absolute and relative frequencies) were computed for all variables for the whole sample and by city. All comparisons between categorical variables were tested with a Pearson χ^2^ test. Second, a multi-level logistic regression model was developed to assess the effects of sociodemographic variables on participation and participants’ PA levels. The multilevel model included two levels: participants of the Ciclovía and study site. All statistical analyses were performed using SAS version 9.2 software (SAS Institute Inc., Cary, NC), Stata Software version 16.0 (StataCorp LLC, College Station, TX, USA), and RStudio (ver. 1.1.453, Rstudio, Inc., Boston, MA, USA) [72].

## 3. Results

### 3.1. Sociodemographic Characteristics

Descriptive characteristics of the study population for each city are presented in Table 3. Approximately, over half of the participants were men (52.29%). The majority of participants were in the 30–49 age group (45.76%), were single (55.45%), had high educational attainment (at least College or technical studies) (49.21%), lived in the middle SES (51.74%), and had a car at home (50.33%).

### 3.2. Health Characteristics

Most of the participants in Bogotá (58.74%) and Santiago de Cali (65.69%) reported their health to be good, and in Santiago de Chile to be excellent (69.08%). On the other hand, most participants in Bogotá, Mexico City, and Santiago de Cali had normal BMI (52.31%) (Table 3). The proportion of the participants in the normal weight category ranged from 44.09% (Mexico City) to 64.34% (Bogotá). Furthermore, an important proportion of participants in these three cities were overweight, this proportion ranged from 29.67% (Bogotá) to 40.47% (Santiago de Cali) (Table 3).

### 3.3. Physical Activity Behaviors

Overall, most of the participants met PA recommendations during the program (51.13%), ranging from 27.18% (Santiago de Chile) to 87.24% (Mexico City) (Table 3). In Bogotá and Mexico City, the majority of Ciclovía Recreativa participants reported meeting the LTPA recommendations (85.61% and 60.19%, respectively), and more than 80% of users in both cities reported meeting overall PA recommendations (85.61% in Bogotá, 83.36% in Mexico City) (Table 3). In contrast, most Ciclovía Recreativa users in Bogotá met PA recommendations by walking and cycling for transport (52.35%), while most users in Mexico City did not (51.04%) (Table 3).

### 3.4. Program Use

Roughly half of the participants reported cycling during the program (52.67%), and to spend less than 3 h in the program (54.42%) (Table 3). The majority of participants were regular users (≥48 events/year) (49.48%) and came alone to the program (62.60%) (Table 3). When participants were asked about motivations to attend the program, the main reasons in Bogotá, Santiago de Cali, and Santiago de Chile included “for recreation or PA” (77.02%, 76.19%, and 60.35%, respectively) and “because of the health benefits” (57.50%, 61.86%, and 22.69%, respectively) (Table 3). Additionally, when participants were asked about the type of activities they normally do when they do not attend the program, most of them reported that they would be engaged in “other types of PA” (64.22%) (Table 3).

### 3.5. Participants’ Perceptions

Regarding participants’ safety perceptions, most of them reported feeling safe with respect to crime in the Ciclovía program (65.51%) (Table 3).

### 3.6. Multi-Level Associations with Meeting Physical Activity Recommendations and Time Participants Spent in the Program

Results from the logistic regression analyses are shown in Table 4. Meeting PA recommendations did not differ by sex, SES and educational level.

### 3.7. Urban Segregation Index

Overall, among the four evaluated cities, Santiago de Chile had the highest SES diversity within its geographic units, and therefore, the lowest level of urban segregation (mean = 0.82 ± 0.13) (Table 1). Additionally, Bogotá was the city with the lowest entropy index levels (mean = 0.06 ± 0.13) and therefore the highest segregation, followed by Mexico City (mean = 0.11 ± 0.13) and Santiago de Cali (mean = 0.16 ± 0.17) (Table 1). In Santiago de Chile, we found a center-periphery pattern distribution, with most of the census zones with higher entropy levels located in the center of the city, and most of the census zones with lower entropy levels were located in the city’s periphery, mainly in the northeastern and southeastern areas (Figure 1). On the other hand, Bogotá, Mexico City, and Santiago de Cali stand out by having large areas of the city concentrating people of a unique SES in an area (low levels of SES diversity). Despite this, the distribution pattern among the three cities differed (Figure 1). Additionally, we found that most of the Ciclovía Recreativa route in the four cities were located in geographic units classified as being highly segregated and segregated (88.61% in Bogotá, 89.76% in Mexico City, 76.31% in Santiago de Cali, and 88.61% in Santiago de Chile) (Table 1) (Figure 1).

### 3.8. Participants’ Trajectories Through the Ciclovía Program

Participants’ trajectories through the Ciclovía program, as well as, geographical distributions of SES within each city are shown in Figure 2. As it is depicted in this figure, by using Ciclovía’s routes participants can reach a higher SES departing from a lower SES, and vice versa, as the Ciclovía program interconnects geographic units belonging to different SES. Unlike Mexico City and Santiago Chile, where the majority of Ciclovía routes are located in geographic units belonging to high SES categories (90.22% and 77.85%, respectively), routes in Bogotá and Santiago de Cali are distributed more broadly throughout the city (Table 1), allowing participants in these two cities to move along a higher variety of SES environments using the Ciclovía route (Figure 2) (Table 5).

### 3.9. Average SES per Traveled Kilometer

In terms of the role of the Ciclovía allowing participants to move through different socioeconomic environments, and its relation to the participant’s traveled distance, we found the highest mobility in Bogotá, followed closely by Santiago de Cali. In these two cities, the maximum SES percentile differences between the neighborhood of origin and the neighborhoods visited as part of the Ciclovía use were 33.58 (*p*-value < 0.001) and 30.38 (*p*-value < 0.001), respectively, indicating that in these two cities, participants were more likely to visit higher or lower SES neighborhoods than their average SES-of-neighborhood origin, providing participants opportunities to move through different socioeconomic environments through the Ciclovía route. Similarly, participants in Bogotá and Santiago de Cali who departed from high SES areas reached higher socioeconomic mobility than participants from other cities, with a maximum SES percentile difference of −25.60 and −17.25, respectively (Table 5). However, participants in Bogotá traveled further to achieved that difference (9.25 km vs. 4.75 km in Santiago de Cali) (Table 5) (Figure 3). By contrast, participants from Mexico City and Santiago de Chile were more likely to stay in geographic units similar to their average SES-of-origin, having a lower overall socioeconomic mobility during leisure time (Figure 3). Particularly, in Santiago de Chile, participants reached areas that were only at a −0.91 lower SES at 6.25 km (Table 5).

## 4. Discussion

This study shows that the Ciclovía Recreativa in Bogotá, Mexico City, Santiago de Cali, and Santiago de Chile can be a socially inclusive program in highly unequal and segregated urban environments. We found that this program provided participants opportunities to move along different socioeconomic environments through the Ciclovía route and to potentially come into contact and potentially interact with people from different socioeconomic conditions during recreational activities. This mobility for recreation was highest in Bogotá and Santiago de Cali, and less prominent in Mexico City and Santiago de Chile. This study underlines the potential of Ciclovía Recreativa programs to transform urban mobility by temporarily changing the streets use from “streets for motorized transport” towards “streets for people”, allowing to democratize and reclaim the streets for recreational and leisure purposes.

Regarding the ability of Ciclovía participants to reach higher or lower SES than their average SES-of-origin using the Ciclovía route, our results suggest that in addition to providing a space for PA, the temporary transformations of the streets for people that takes place during the Ciclovía events had complimentary benefits such as getting citizens to share public space and connecting people from diverse communities. In such a context, Ciclovía works as a venue that facilitates physical proximity, exposure to new communities, and interactions between different socioeconomic groups while promoting diversified spaces, all essential factors for promoting social integration [73]. We hypothesized that the mechanism through this happens as follows: the program provides a public space within the urban context for the integration and coexistence between individuals from diverse SES; the willingness of participants to have a greater level of social contact, as well as, the desire to participate in civic and social activities; and the value that users give to the existing opportunities for contact. These are supported by the fact that none of the routes that Ciclovía users embark on are compulsory, but rather they are chosen purely for recreational purposes, suggesting that by temporarily removing the danger of motor vehicles, the Ciclovía provides a novel type of public space that could help people make social connections by increasing opportunities for meeting people from varied socioeconomic conditions.

Recent research conducted in the region about the effects that spatial proximity and social contact among people from different socioeconomic levels have on social cohesion has shown that a society offering higher and valued contact opportunities among different groups would show a tendency to be more cohesive in a less conflictive context, so long as people take such opportunities [19,74]. Accordingly, the Ciclovía program could potentially improve social cohesion, referred to as the individual’s disposition to construct coexistence patterns with ‘others’ whose social conditions may be different [74]. However, given the temporality nature of the Ciclovía program, this effect could be transient, as well as, it could be higher in Santiago the Cali, where Ciclovía events take place at least twice a week. These results are relevant for policymakers and stakeholders who should consider implementing Ciclovía routes as a means to promote social cohesion. Particularly, in Mexico City and Santiago de Chile, routes should be implemented in places with lower SES, as our results showed that Ciclovías routes in these two cities are concentered in geographic units with higher SES. This will not only allow individuals from lower SES to travel lower distances to enjoy the Ciclovía program, but also increase the chances to coincide and interact and coincide with individuals from varied SES.

We also found that people that attended the Ciclovía program have a higher perception of security: 65.51% of Ciclovía participants reported feeling safe at the program, despite the generalized insecurity perception participants had in their respective cities. According to a recent citywide survey, 84% of Bogotá [75], 61.8% of Santiago de Cali [75], and 76.8% of Santiago de Chile [76] residents reported feeling unsafe in the city, suggesting that the Ciclovía program could potentially enhance the interpersonal trust of individuals and allow them to feel safer by reducing crime perception [77]. Possible explanations of how Ciclovía Recreativa improved participant’s perception of security included the presence of more pedestrians, as well as people engaging in positive activities such as PA in the program [77]. Furthermore, these results suggest that the Ciclovía has the potential to improve social capital, particularly, the social trust (e.g., generalized trust, interpersonal trust) and social networks (e.g., informal relationships) dimensions of it [78]. Social capital refers to those social resources that may be accessed across groups of different socioeconomic or sociodemographic characteristics, and its presence helps to build trust and maintain channels of communication between individuals [79]. A program with the potential capacity of promoting social cohesion and social capital on a weekly basis is relevant as some studies have positively associated these two factors with well-being [36,79,80,81,82].

The evidence generated about the potential role of the Ciclovía program helping to mitigate the existing patterns of urban segregation has great implications for decision-makers, who should prioritize this massive community-wide intervention as a means of targeting social inclusion, which could secondarily lead to improvements in social cohesion and social capital, and ultimately result in broad-ranging effects on the well-being of citizens. Further studies should evaluate and document the impact that the Ciclovía has on improving social capital and social cohesion, like monitoring, publishing, and publicly debating the effects of this program is an essential ingredient for its longer-term sustainability [29].

Another key finding of our study was the high prevalence of participants meeting PA recommendations during the Ciclovía program (51.13%), as well as the important proportion of participants that achieved at least 60% of the weekly PA recommendations by attending the program (54.42%). It is also important to underscore that meeting PA recommendations in the Ciclovía did not differ by gender or SES. This is especially important in Latin America and the Caribbean Region, the region with the overall highest prevalence of insufficient PA worldwide, particularly among women [41]. Therefore, the Ciclovía program could play a vital role in providing spaces at the community level aimed at promoting healthy habits, in cities with limited recreational public spaces to encourage PA practice. Programs such as the Ciclovía has the potential of working as an effective, scalable, and community-based intervention that facilitates regular PA, as well as, could improve health across diverse populations.

Regarding the Ciclovía program fostering healthy habits, our results suggested that people who attended the Ciclovía have a higher perceived health status: approximately 65% of Ciclovía participants perceived their health condition as either ‘Excellent’ or ‘Good’. Moreover, among the main reasons for participating in the program, as reported by users, the health benefits associated with PA were at the top (74.03%). The promotion of PA often leads to the adoption of other healthy lifestyle behaviors and is associated with numerous behavioral and emotional improvements [83,84,85]. In this context, the Ciclovía program could be used to generate public health interventions on programs that have been implemented and use them as a bridge for motivating people for people from different SES to adopt healthy habits. Specifically, in Bogotá, the Health Secretary developed the program “Cuídate, se feliz” [86], in which a team consisting of nurses, nutritionists, physical therapists, and physical educators, offers free health assessments (i.e., blood pressure measurements, anthropometric, and nutrition evaluations) at different points of the public space, such as parks, shopping centers, bike routes, public transport stations, and the Ciclovía program, during street market-like events. Once the data is gathered, the user’s information is submitted in a system that characterizes and identifies patients at risk of unfavorable outcomes, who ultimately will be directed by the team professionals to the health-care network.

This study is consistent with findings of previous studies, including that Ciclovía encourages people to be physically active, as well as that it has the potential of promoting social cohesion and social capital. In a study carried out in 2013 that evaluated the effect of San Diego’s Open Streets program (CicloSDias) on PA, researchers found that 97% of participants met the 30 min/day guideline, 39% met the 150 min/week guideline, and 27% would have been sedentary without event [87]. Similarly, a cross-sectional study conducted in 2012 found that the majority of Ciclovía participants in Bogotá met LTPA recommendations. Additionally, this study found that Ciclovía participants in Bogotá reported having higher social capital levels across three evaluated aspects (shared values, trust, and willingness to help each other). Authors found that 62.4% of participants reported willingness to help each other and 61.4% to get along with each other, 51.2% reported to feel safe at the event with respect to traffic and accidents, and 42.4% with respect to crime [36]. In a study conducted between 2008 and 2009 that evaluated the Ciclovía program in Chicago, researchers found that individuals identified contact with neighbors and exposure to new communities as the primary benefit of participation in the program. Their results suggested that the program in Chicago promoted inter-community organizational partnerships, community organizing, and resident interaction [88]. In a recent systematic review of different city street experiments around the world that transform urban mobility, including the Ciclovía program, it was highlighted the sizable positive impacts of Ciclovía Recreativa programs on promoting PA, enabling a modal shift from car to walking, cycling, and public transport, improving safety and enhancing social interaction and social capital [29]. Particularly, the review showed the potential effect these programs have on providing a platform for the development of collaborative relationships, both between local organizations and with residents, as well as, to be exposed to other communities [29].

Finally, taking advantage of the temporal nature of the Ciclovía program, as well as, its capability of enhancing the public space function of city streets, over the past few months, urban planners have explored how to apply tactical urbanism measures as a way to respond to all the mobility challenges that have emerged to the COVID-19 pandemic. Tactical urbanism measures are low-cost, scalable, problem-solving responses in the short term aimed at promoting behavioral changes in urban dwellers through innovative interventions [89]. Governments worldwide have found utility in the idea that they can act quickly, and make a change to their street environment [90]. For instance, in July 2020, at least 92 cities in 20 countries on 3 continents had expanded sidewalks and bike-lanes width, length, and connectivity had implemented new everyday Ciclovía programs or expanded their current programs, as strategies that support physical distancing and traffic safety. In fact, three out of four cities included in this study—Bogotá [91], Mexico City [92], and Santiago de Chile [93]—were among the first cities worldwide that added to their existing routes temporary “pop-up” cycle lines. In Bogotá, the city’s mayor expanded the 550 km of existing permanent bike lanes adding 80 km of new lanes, 35 km in mid-March, and 45 km in mid-April, for a total of 630 km of bike lanes functioning 24 h a day, the seven days a week. This is an example of how Ciclovía programs should be seen as potential emergency resources that can be quickly adaptable according to citizen’s needs and hopefully, they will become permanent after the post-pandemic period.

### Limitations and Strengths

This study has strengths and limitations. First, the sample size included in this study is the largest to date in a study aimed at evaluating the Ciclovía program. Furthermore, this is the first study to our knowledge to make a comparative multicounty analysis of four Ciclovía Recreativa programs in Latin America using a dynamic urban segregation index for measuring social inclusion. Unlike studies that used traditional static segregation measures, we employed an objective dynamic measurement that allowed us to better understand the potential role of the Ciclovía program on social inclusion. Despite the rigorous harmonization of the data, the four studies used questionnaires that were previously adapted for each city. Therefore, future studies should consider including diverse cities with the same standardized questionnaires to have better comparability of indicators. Second, using GIS for estimating participant’s trajectories throughout the program assumed routes that did not necessarily reflect those actually taken for program participants. Despite this, modeling routes with GIS has been recommended as an acceptable method for trajectories prediction/estimation, particularly for active commuting [94]. Future studies should combine GPS and GIS tools to record the participant’s trajectories in the program, which would enhance the accuracy of Ciclovías participant’s trajectories [95,96,97]. Third, SES proxies and geographical units employed were different across study sites, which we expect added some error to analyses and reduced observed associations. However, we harmonized and developed a data structure that accommodates the best available proxies in each city at the lowest available geographic units, allowing us to have comparable data across the four cities included in the study. This process was guided by the principle that pragmatic albeit imperfect definitions would be necessary to optimize the use of available data and advance in the knowledge of the potential benefits of the Ciclovías. Fourth, to date, there is no theoretical or empirical basis for determining the optimal cut-points in categorizing the entropy index, we categorized this continuous variable into quantiles. However, by doing this, we were able to characterize the distribution of entropy in metropolitan areas. Further research may be needed to determine the best cut-off points for defining categories for which this spatial segregation measurement has positive or negative effects on urban segregation in the Latin American context.

## 5. Conclusions

The Ciclovía Recreativa program, as implemented in four Latin American cities (Bogotá, Mexico City, Santiago de Cali, and Santiago de Chile), can be a socially inclusive program in highly unequal and segregated urban environments. Our results suggested that, in addition to providing a space for PA, the Ciclovía program works as a venue that facilitates physical proximity, exposure to new communities, and interactions between different socioeconomic groups, having complimentary benefits such as getting citizens to share public space and potentially promoting social cohesion and social capital. This study illustrates the importance of implementing public space usage policies, as well as built environment changes in urban settings, to have a population-based impact in aspects of public health, such as social inclusion, equity, and PA.

## Figures and Tables

**Figure 1 ijerph-18-00655-f001:**
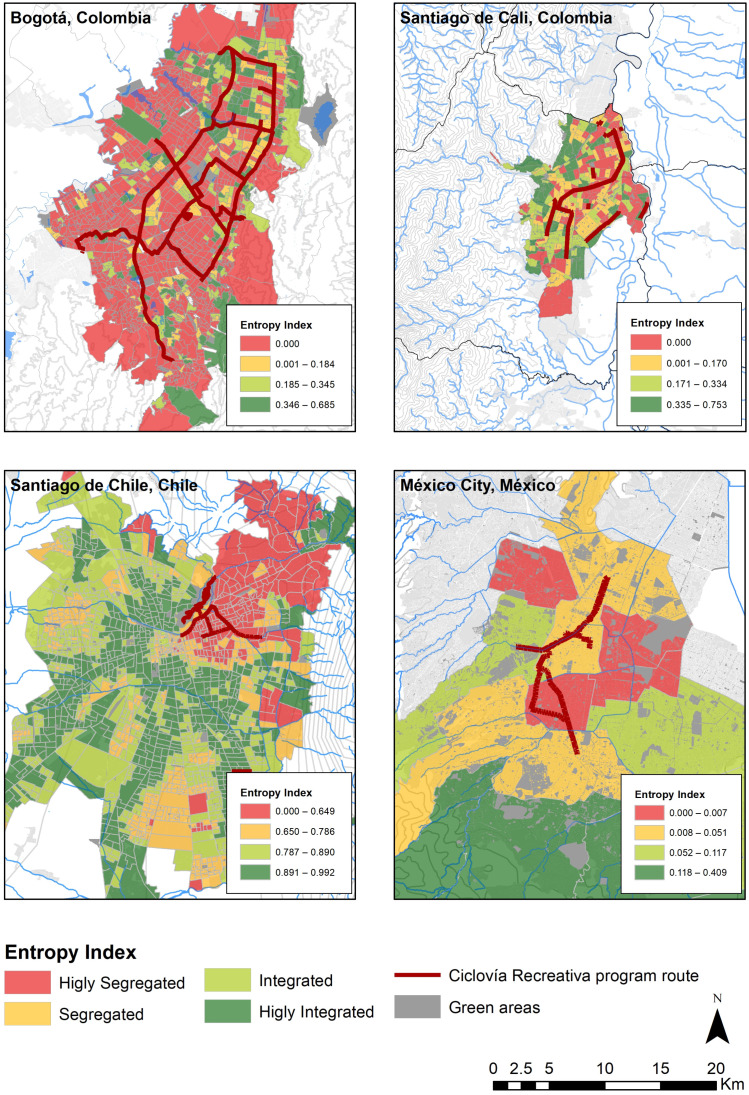
Geographic distributions of local spatial entropy index within the four cities included in the study. Ciclovía Recreativa program routes and green areas in each city are also shown.

**Figure 2 ijerph-18-00655-f002:**
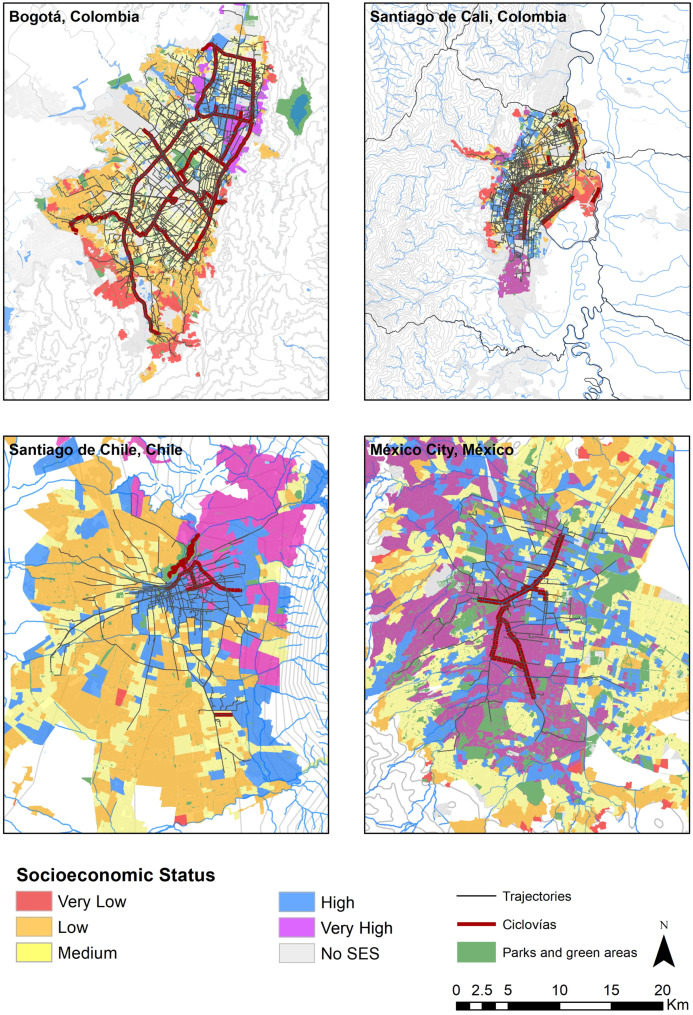
Participants trajectories through the Ciclovía program. Geographical distributions of Socioeconomic Status, Ciclovía Recreativa program routes, and green areas within each city are also shown.

**Figure 3 ijerph-18-00655-f003:**
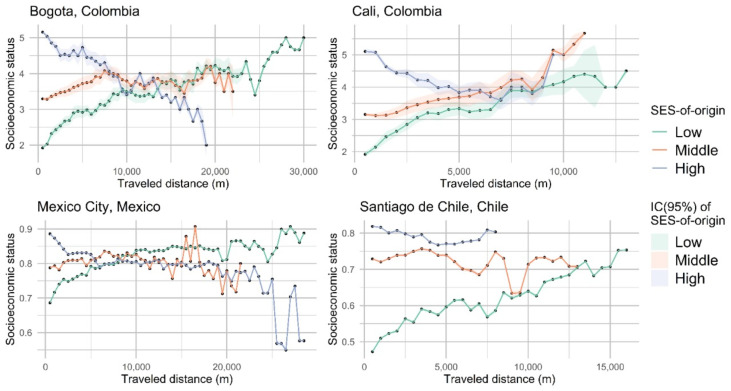
Maximum SES percentile difference reached by participants using the Ciclovía route, compared with their average SES-of-origin, as well as distance participants traveled to achieve that difference; SES: socioeconomic status.

**Table 1 ijerph-18-00655-t001:** City and Ciclovía Recreativa program characteristics by study site.

Characteristic	Bogotá	Mexico City	Santiago de Cali	Santiago de Chile
**City Characteristic**				
Population metrics				
Total population ^a^	7,878,783	8,918,653	2,470,852	7,112,808
Population Density ^b^	21,916	11,247	17,691	9632
Inequality metric				
GINI coefficient	0.50	0.53	0.46	0.49
Urban segregation metrics				
Entropy index (Mean ± SD)	0.06 (0.13)	0.11 (0.13)	0.16 (0.17)	0.82 (0.13)
Crime metric				
Homicide rate ^c^	14.30	16.00	51.30	4.90
Urban landscape metrics				
Patch density ^d^	0.71	0.64	1.06	0.66
Green area per capita ^e^	3.90	5.40	5.93	4.83
Transportation metrics				
Motorization rate ^f^	247.00	544.05	251.28	254.67
Urban travel delay index ^g^	0.82	0.57	0.57	0.34
**Ciclovía program characteristics**				
Name	Ciclovía de Bogotá	Muévete En Bici	La Ciclovida de Cali	CicloRecreoVía
Year of inauguration	1974	2007	1996	2006
Length (Km)	127.69	55	60	38
Schedule	Su- Ho 7 h	Su 6 h	Su- Ho 5 h & Th 2 h	Su 4 h
Participants per event	600,000–1,750,000	21,000	30,000	40,000
Events per year	66–72	37	93	51
Source of funding	Public and Private	Public	Public	Public and Private
Program average cost per year (USD millions) ^h^	2.40	1.00	1.65	1.06
Number of independent circuits	1	1	7	9
Scale	Metropolitan	Metropolitan	Metropolitan	Metropolitan
Percentage of the Ciclovía route in				
Low SES	18.04	0	23.71	15.37
Middle SES	61.64	9.78	64.72	6.78
High SES	20.32	90.22	11.58	77.85
Percentage of the Ciclovía route in				
Highly segregated geographical units	66.03	27.29	21.18	61.54
Segregated geographical units	10.28	62.47	32.3	27.07
Integrated geographical units	11.67	10.24	25.97	--
Highly integrated geographical units	12.02	--	20.55	11.39
**Physical activity levels**				
Meeting physical activity recommendations (%) ^i^	66.00	71.10	66.00	73.40%
Meeting physical activity recommendations (%) ^i^, males	61.20	74.50	61.20	75.60%
Meeting physical activity recommendations (%) ^i^, females	51.10	67.80	51.10	71.40%

Th: Thursday, Su: Sunday, Ho: holidays, and h: hours. SES: socioeconomic status. ^a^ Total population: total population within the geographic boundary. ^b^ Population Density: population per hectare of all the built-up area inside the geographic boundary. ^c^ Homicide rate: Victims of intentional homicide per 100,000 population. ^d^ Patch density: measures the extent to which urban settlements (or patches) are close together (aggregated) or dispersed (fragmented). ^e^ Green area per capita: area of green spaces per capita in m^2. f^ Motorization rate: measures the number of total registered motorized vehicles per 1000 inhabitants per geographic unit and year. ^g^ Urban travel delay index: measures the increase in travel times due to congestion in the street network. ^h^ Program average cost in millions: average cost of the program for the most recent year in millions (USD). ^i^ Percentage of adults aged 18–64 years old who do at least 150 min of moderate-intensity, or 75 min of vigorous-intensity physical activity per week, or any equivalent combination of the two, at the national level. Estimates in the three countries captured physical activity across all domains of life, including work/household, transport, and leisure time.

**Table 2 ijerph-18-00655-t002:** Origin and departure points by study cite.

Study Site	Origin Point	Destination Point
Bogotá	Participant’s home address or the nearest intersection to the participant’s home addresses.	The address of the place or the nearest intersection where participants were interviewed.
Mexico City	The nearest intersection to the place participants started their journey.	The destination point was the nearest intersection to the farthest point participants intended to reach during their journey.
Santiago de Cali	The nearest intersection to the participant’s home addresses.	The nearest intersection where participants were interviewed.
Santiago de Chile		

**Table 3 ijerph-18-00655-t003:** Sociodemographic, health, program use and physical activity characteristics of the Ciclovía Recreativa program participants by study site.

Characteristic	Bogotá			Mexico City			Santiago de Cali			Santiago de Chile			Overall			Multi-Variable
	N = 1001			N = 721			N = 1159			N = 401			N = 3282			
	n	%	* *p*-Value	n	%	* *p*-Value	n	%	* *p*-Value	n	%	* *p*-Value	n	%	* *p*-Value	* *p*-Value
**Sociodemographic characteristics**																
**Sex**																
Male	619	61.84%	<0.001	370	51.32%	0.479	583	50.48%	0.746	142	35.41%	<0.001	1714	52.29%	0.009	<0.001
Female	382	38.16%		351	48.68%		572	49.52%		259	64.59%		1564	47.71%		
**Age group (years)**																
18–29	367	36.74%	<0.001	255	35.61%	<0.001	306	26.40%	<0.001	115	30.42%	<0.001	1043	32.07%	<0.001	<0.001
30–49	451	45.15%		334	46.65%		522	45.04%		181	47.88%		1488	45.76%		
≥50	181	18.12%		127	17.74%		331	28.56%		82	21.69%		721	22.17%		
**Marital status ^a^**																
Single	574	57.46%	<0.001	458	63.61%	<0.001	563	48.58%	0.332	--	--	--	1595	55.45%	<0.001	<0.001
Living with a partner	425	42.54%		262	36.39%		596	51.42%		--	--		1283	44.55%		
**Highest level of educational attainment**																
Primary	44	4.42%	<0.001	8	1.11%	<0.001	44	3.83%	<0.001	1	0.29%	<0.001	97	3.03%	<0.001	<0.001
Secondary/High school	287	28.82%		272	37.73%		419	36.50%		71	20.94%		1049	32.74%		
College//technical	535	53.71%		363	50.35%		617	53.75%		267	78.76%		1782	55.65%		
Master’s degree or higher	130	13.05%		78	10.82%		68	5.92%		0	0.00%		276	8.61%		
**Socioeconomic Status**																
Low	207	20.68%	<0.001	4	0.61%	<0.001	335	28.90%	<0.001	32	9.44%	<0.001	578	18.31%	<0.001	<0.001
Middle	655	65.43%		52	7.90%		751	64.80%		264	77.88%		1722	54.55%		
High	139	13.89%		602	91.49%		73	6.30%		43	12.68%		857	27.15%		
**Car owner in household**																
Yes	458	45.75%	0.007	202	36.66%	<0.001	703	60.81%	<0.001	--	--	--	1363	50.33%	0.729	<0.001
No	543	54.25%		349	63.34%		453	39.19%		--	--		1345	49.67%		
**Health characteristics**																
**Perceived health status**																
Excellent	314	31.37%	<0.001	--	--	--	230	19.88%	<0.001	277	69.08%	<0.001	821	32.08%	<0.001	<0.001
Good	588	58.74%		--	--		760	65.69%		104	25.94%		1452	34.86%		
Fair/Bad/Poor	99	9.89%		--	--		167	14.43%		20	4.99%		286	33.06%		
**Body Mass Index category ^b^**																
Underweight	20	2.00%	<0.001	4	0.55%	<0.001	11	0.95%	<0.001	--	--	--	35	1.21%	<0.001	<0.001
Normal weight	644	64.34%		352	48.82%		511	44.09%		--	--		1507	52.31%		
Overweight	297	29.67%		270	37.45%		469	40.47%		--	--		1036	35.96%		
Obese	40	4.00%		95	13.18%		168	14.50%		--	--		303	10.52%		
**Physical activity characteristics**																
**Meeting PA recommendations during the Ciclovía ^c^**																
Yes	453	45.25%	0.003	629	87.24%	<0.001	487	42.02%	<0.001	109	27.18%	<0.001	1678	51.13%	0.197	<0.001
No	548	54.75%		92	12.76%		672	57.98%		292	72.82%		1604	48.87%		
**Meeting weekly LTPA recommendations ^c,d^**																
Yes	688	68.73%	<0.001	434	60.19%	<0.001	--	--	--	--	--	--	1122	65.16%	<0.001	<0.001
No	313	31.27%		287	39.81%		--	--		--	--		600	34.84%		
**Meeting weekly overall PA recommendations (transportation or leisure) ^c^**																
Yes	857	85.61%	<0.001	601	83.36%	<0.001	--	--	--	--	--	--	1458	84.67%	<0.001	0.200
No	144	14.39%		120	16.64%		--	--		--	--		264	15.33%		
**Meeting PA recommendations (transport) ^c^**																
Yes	524	52.35%	0.137	353	48.96%	0.576	--	--	--	--	--	--	877	50.93%	0.441	0.165
No	477	47.65%		368	51.04%		--	--		--	--		845	49.07%		
**Program use characteristics**																
**Type of activity in the Ciclovía**																
Cycling	603	60.36%	<0.001	634	87.93%	<0.001	205	18.00%	<0.001	275	68.58%	<0.001	1717	52.67%	<0.001	<0.001
Rollerblading	52	5.21%		25	3.47%		21	1.84%		29	7.23%		127	3.90%		
Walking	222	22.22%		23	3.19%		464	40.74%		16	3.99%		725	22.24%		
Running/jogging	118	11.81%		38	5.27%		163	14.31%		73	18.20%		392	12.02%		
Other ^e^	4	0.40%		1	0.14%		286	25.11%		8	2.00%		299	9.17%		
**Time spent in the program (h)**																
<3 h	586	58.54%	<0.001	92	12.76%	<0.001	796	68.68%	<0.001	312	77.81%	<0.001	1786	54.42%	<0.001	<0.001
3–4 h	210	20.98%		279	38.70%		231	19.93%		62	15.46%		782	23.83%		
≥4 h	205	20.48%		350	48.54%		132	11.39%		27	6.73%		714	21.76%		
**Frequency of participation (Events) ^f^**																
At least once a year	68	6.81%	<0.001	127	17.61%	<0.001	110	9.52%	<0.001	184	45.89%	<0.001	489	14.93%	<0.001	<0.001
Once per month	120	12.01%		95	13.18%		105	9.09%		59	14.71%		379	11.57%		
Two/Three times per month	258	25.83%		282	39.11%		200	17.32%		47	11.72%		787	24.02%		
≥ 4 times per month	553	55.36%		217	30.10%		740	64.07%		111	27.68%		1621	49.48%		
**People accompanied by at the program ^g^**																
Came alone	567	56.64%	<0.001	473	65.69%	<0.001	763	65.83%	<0.001	--	--	--	1803	62.60%	<0.001	<0.001
Came with another person	434	43.36%		247	34.31%		396	34.17%		--	--		1077	37.40%		
**Reasons to attend the program**																
Share with family/friends	324	32.37%	<0.001	--	--	--	484	41.76%	<0.001	35	8.73%	<0.001	843	32.92%	<0.001	<0.001
Used for recreation/PA	771	77.02%	<0.001	--	--	--	883	76.19%	<0.001	242	60.35%	<0.001	1896	74.03%	<0.001	<0.001
Health Benefits	575	57.50%	<0.001	--	--	--	717	61.86%	<0.001	91	22.69%	<0.001	1383	54.02%	<0.001	<0.001
Other ^h^	144	14.40%	<0.001	--	--	--	213	18.38%	<0.001	133	33.17%	<0.001	490	19.14%	<0.001	<0.001
**Activities if participants were not in the Ciclovía**																
Stay at home	93	9.36%	<0.001	149	21.07%	<0.001	276	24.30%	<0.001	--	--	--	518	18.26%	<0.001	<0.001
Sedentary activity	94	9.46%		81	11.46%		190	16.76%		--	--		365	12.87%		
Other type of physical activity	701	70.52%		477	67.47%		644	56.69%		--	--		1822	64.22%		
Other ^i^	106	10.66%		--	--		26	2.29%		--	--		132	4.65%		
**Participants’ perceptions at the program**																
**Safety perception (crime)**																
Unsafe	37	3.70%	<0.001	--	--	--	494	42.62%	<0.001	22	5.49%	<0.001	553	21.60%	<0.001	<0.001
Safe	915	91.50%		--	--		383	33.05%		379	94.51%		1677	65.51%		
Neither safe nor unsafe	48	4.80%		--	--		282	24.33%		--	--		330	12.89%		

* Comparisons between categorical variables were tested with a Pearson χ^2^ test. ^a^ Marital status: Partner—partner or married. Single—widowed, divorced, separated. ^b^ Body Mass Index category (kg/m^2^)- Underweight (<18.5); Normal weight (18.5–24.9); Overweight (25.0–29.9); Obese (≥30.0). ^c^ ≥150 min/week of moderate-vigorous physical activity. ^d^ Meeting weekly physical activity recommendations during leisure time. ^e^ Type of physical activity at the program: Other- combination of different physical activities, skateboarding, free exercise classes (aerobics), rollerblading, pushing the baby strollers/wheelchair. ^f^ Frequency of participation: At least once a year (1–11 events/year); Once per month (12–23 events/year); Two/Three times per month (24–47 events/year); Four or more times per month (≥48 events/year). ^g^ People accompanied by at the program: Came with another person—spouse/partner, family (children, siblings, cousins, father/mother), co-workers, neighbors, classmates, friends. Came alone-came alone/pets. ^h^ Reasons to attend the program: Other—combination of previous options, to protect the environment, to attend to free exercise classes (aerobics), because it is free, because I do not have anything else to do, for tourism, to get to know the city, to avoid robberies, because they feel safer, to do diligences, it is a habit, to walk their pet, to meditate, to think or take the sun, for transportation, to learn how to dance, because it is given by the mayoralty, to prepare for another activity, for working. ^i^ Activities if participants do not attend the program: Other-combination of previous options, housekeeping duties, working, drinking alcohol, to go to church, to go to the cinema, to travel, to study, to work.

**Table 4 ijerph-18-00655-t004:** Odds ratios of meeting the PA recommendations from WHO associated with participant’s socioeconomic status and the highest level of educational attainment.

		Multivariable		
Independent Variable †		Odds Ratio	95% CI	*p*-Value
Meeting physical activity recommendations during the Ciclovía †				
Covariates	Sex	0.90	[0.54; 1.49]	0.684
Socio-economic status				
	Low	Reference group		
	Middle	0.94	[0.27; 3.23]	0.928
	High	1.21	[0.66; 2.18]	0.540
Highest level of educational attainment				
	Primary	Reference group		
	Secondary/High school	0.92	[0.32; 2.57]	0.875
	College/technical	1.43	[0.51; 3.94]	0.490
	Master’s degree or higher	1.88	[0.63; 5.59]	0.258

† Model includes the city as random effects; SES: socioeconomic status; WHO: World Health Organization.

**Table 5 ijerph-18-00655-t005:** Participants’ trajectories analyses by study site.

	Bogotá		Mexico City		Santiago de Cali		Santiago de Chile		Overall
Overall	%	*p*-Value	%	*p*-Value	%	*p*-Value	%	*p*-Value	*p*-Value
**Average percentage of the participants’ trajectories in**									
Low SES	15.13	<0.001	0.24	<0.001	16.20	<0.001	10.23	<0.001	<0.001
Middle SES	66.24		13.88		68.55		77.46		<0.001
High SES	18.63		85.89		15.26		12.31		<0.001
**Maximum difference (percentile) by SES-of-origin †**									
Low SES	33.58	<0.001	13.84	<0.001	30.38	<0.001	15.06	<0.001	<0.001
Middle SES	8.94		1.55		16.22		−0.91		<0.001
High SES	−25.60		−11.06		−17.25		−2.76		<0.001
**Average distance traveled (km) by SES origin**									
Low SES	14.75	0.004	14.00	0.060	6.25	0.377	7.75	0.013	<0.001
Middle SES	10.75		10.50		5.25		6.25		<0.001
High SES	9.25		14.00		4.75		3.75		<0.001

† Maximum SES percentile difference reached by participants using the Ciclovía route, compared with their average SES-of-origin; SES: socioeconomic status.

## Data Availability

The data presented in this study are available on request from the corresponding author. The data are not publicly available due to privacy restrictions.

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
