# Peer review of "Social Inclusion and Physical Activity in Ciclovía Recreativa Programs in Latin America"

_ijerph, 2021, doi:10.3390/ijerph18020655_

Round 1

Reviewer 1 Report

The paper studied the Ciclocia Recreativa programs in four Latin American cities. It is a very important topic and the authors carried out the extensive analysis. The paper significantly improved the literature on Latin American studies. The following details need to be corrected before publication.

  • Line 73 to line 78 the format of references need to be corrected
  • The citations named as Author et.al. Need to be corrected
  • Line 104 needs to clarify, “Ciclovía is the largest program in the world.”
  • Line 108 citations need to be corrected
  • Line 113 citations need to be corrected
  • Line 128 citation need to be corrected
  • Line 231 needs to clarify r means
  • Suggest to put the four city maps into one map framework and unify the scale and legend in figure 1 and 2, current map colours not the same and cross multi-page,
  • Figure 3 needs a better resolution and darker grid references

Reviewer 2 Report

This paper mainly shows that the Ciclovía Recreativa in Bogotá, Mexico City, Santiago de Cali, and Santiago de Chile can be a socially inclusive program in highly unequal and segregated urban environments.

The introduction is logical and progressive, and the research problem is clear. The article begins by describing the current situation of urban segregation in Latin America, followed by stating the innovative public space and transport programs. Then, relevant research was described, thus leading to the research problem in this paper. Finally, define how to measure the segregation and propose the purpose of this paper, namely, to compare participant’s spatial trajectories in four Ciclovía Recreativa programs in Latin America according to socioeconomic characteristics and urban segregation of these cities; and secondly, to assess the relationship between participants´ PA levels and sociodemographic characteristics.

However, there are still some problems I think need further modification. At the end of the paper, the author also mentioned the limitations of this study. Firstly, future studies should combine GPS and GIS tools to record the participant’s trajectories in the program to enhance the accuracy of Ciclovías participant's trajectories. Secondly, SES proxies and geographical units employed were different across study sites, which we expect added some error to analyses and reduced observed associations. And I think this paper lacks a large number of relevant literature review, thus lacking theoretical or empirical basis.

Taken as a whole, this is a practical paper that illustrates the importance of implementing public space usage policies, as well as built environment changes in urban settings, to have a population-based impact in aspects of public health, such as social inclusion, equity, and PA.

Reviewer 3 Report

This is a valuable piece of research that tackles the reality of a wonderful initiative in Latin American cities which is Ciclovia. The manuscript is well written and structured, with up to date references. However in my opinion the manuscript is not solid in its methodological design and its psychometric properties. Here we stress, in our opinion, some limitations:

  • Topic presentation in the introduction is scarce. A deeper introduction is needed in order to frame the study aim, with concepts and variables that can frame Ciclovia as an unique event. Rates of PA in cities studied are not provided, as an indicator of the potential use of the bike. Also, which type of PA are performed mainly along Ciclovia? (cycling, running, skating, etc). This are two questions that will enhance the application and concretion of the study, and both may have an influence in data interpretation.
  • Please revise spaces between words along the text: issues with them we saw in lines 51, 53, 77, 86, etc. Please apply to thw hwole text.
  • Please rephrase line 104 “Ciclovía is the largest program in the world”: it´s too general.
  • We realized that authors were using different questionnaires and tools for data collection, and sometimes merging data from different sources. This is very confusing. Also for instance, PA levels are measured using different scales by city, and no information about this dimension in another two cities.
  • No information about how research staff was trained to interview the participants is provided.
  • The main limitation I find of this study refers to the design and validation of the questionnaire / interview used. No information about validation, feasibility or reliability is provided.
  • In opinion of this reviewer, the manuscript presents strong methodological limitations that limit study reproducibility that could affect data interpretation.

Because its novelty and appropriateness, we suggest authors to explain in more detail the prior issues, as we truly believe the topic deserves this kind of studies.
